# Blessing or Curse? The Impact of Digital Technologies on Carbon Efficiency in the Agricultural Sector of China

**Yong Zhu [1], Xiongying Wang [2] and Gong Zheng [3,*]**

[1] School of Business, Hunan University of Science and Technology, Xiangtan 411201, China; zhuyong@mail.hnust.edu.cn
[2] School of Business, Hunan Women's University, Changsha 410004, China; 160115030003@mail.hnust.edu.cn
[3] Teaching and Research Department, Shanghai National Accounting Institute, Shanghai 201702, China
[*] Correspondence: zhenggong@snai.edu

**Abstract:** Digital technology can be used to adjust the structure of energy production, promote the development of new agricultural production technologies, and reduce carbon emissions. With the increasing integration of digital technology in various fields, rural development is also entering a critical period of digital transformation. Therefore, this paper uses digital technology and agricultural carbon emission intensity as research objects. We use panel data from 2011 to 2019. We first measure and analyse the level of digital technology development in China. This article empirically tests the impact between digital technology and the intensity of agricultural carbon emissions. Digital technology can be used to significantly suppress the intensity of agricultural carbon emissions. The suppressive effect is more significant in the western region of China than in the central and eastern regions.

**Keywords:** digital economy; agricultural carbon emissions; carbon reduction; emerging economies

## 1. Introduction

Climate change has become an enormous problem and has caused tremendous pressure on mankind. Although China has made significant progress since its reform and opening up, its carbon dioxide emissions have been increasing rapidly, which has accompanied China's rapid economic development. Data published by the World Resources Institute (WRI) show that China's total carbon emissions rank first globally. According to World Energy Statistics Yearbook 2021 statistics, in 2020, the world's total carbon emissions was 32.248 billion tonnes. China's total carbon emissions was 9.90 billion tonnes, accounting for 30.7% of the global share. Global warming is a concern for everyone worldwide, so governments are taking active measures, and China is no exception. China said at a conference on global warming in September 2020 that it would reach its highest level of carbon dioxide emissions by 2030, achieving zero carbon dioxide emissions by 2060. China's targets for carbon dioxide emission reduction and carbon dioxide neutralisation show that China has made emission reduction a top priority and said that it would follow the concept of sustainable development to slow global warming. As the foundation of the national economy, agriculture also produces many greenhouse gas emissions. Studies show that China's agricultural production accounts for 17% of the country's total greenhouse gas emissions, and the agricultural industry is under tremendous pressure to reduce emissions. Therefore, reducing greenhouse gas emissions, which has become an essential collective responsibility for humans to develop, is vital to reducing agricultural greenhouse gas emissions.

As the importance of the digital economy to China's economic development has become increasingly prominent, how to use digital means to improve the efficiency of reducing greenhouse gas emissions of agricultural products has become the current research focus. Digital technologies can affect carbon emissions (Tang et al., 2019) [1]. Therefore, digital technological advances can help to effectively reduce carbon emissions by reducing "high-carbon" inputs and improving the efficiency of production factors. In agriculture,

digital technological progress can improve resource use and agricultural productivity efficiency, with significant emission-reducing effects. The digital economy cannot be separated from technological innovation (Guo et al., 2023) [2]. Currently, the world is moving into the era of a digital economy, mainly featuring IOT, big data, blockchain, AI (artificial intelligence), cloud computing, and others, and data have become a key factor in production (Guo and Zhong, 2023) [3]. The global digital economy will reach USD 32.6 trillion, representing 43.7% of GDP. The U.S. digital economy is the largest in the world, with a size of USD 13.6 trillion, and China represents USD 5.4 trillion. With continual improvements in digital infrastructure in rural regions, digital elements are penetrating further into agriculture, bringing new opportunities for the high-quality development of the agricultural sector. However, there are fewer studies on the topic of agriculture, among which Tian and Guan (2023) [4] explored the impact and mechanisms of the topic of food cultivation. The digital economy can contribute to reduce carbon emissions from food cultivation. Much of the existing research emphasises the integration of a particular production segment of agriculture with technology, and there is a lack of research on the digital economy and carbon emissions.

The data come from 2011 to 2019. The econometric analysis verifies the impact of digital science and technology on China's agricultural carbon emissions. Section 2 presents a review and summary of the relevant research results. The third section presents an entropy analysis of information technology development in 30 regions of China from 2011 to 2019. The fourth section presents the data-driven empirical analysis of the carbon emission intensity of agricultural products in China and presents the theoretical and empirical analysis. The fifth section presents the conclusion, policy recommendations, and forecasts future research.

## 2. Literature Review

### 2.1. Studies on Agricultural Carbon Emissions

2.1.1. Data Determinant of Agricultural Carbon Emissions

The complexity of agricultural activities, the diversity of sources, and the intersection of the components of carbon emissions increase the difficulty in the accurate assessment of its carbon emissions. The results show that agriculture is the primary source of global carbon emissions, and about 1A5 comes from agriculture (Paustian et al., 1998) [5]. Previous studies have classified and measured them according to different agricultural carbon emission sources. The first aspect is mainly discussed in terms of the consumption of agricultural materials and the disposal of agricultural waste. For example, Johnson et al. (2007) [6] considered that it mainly comes from the application of chemical fertilisers, pesticides, and energy, as well as the improper incineration and landfill of wastes, crops, and straw. The second aspect is based on methane in paddy fields, and carbon dioxide and nitrous oxide in the soil. For instance, calculations by MacLeod (2010) [7] showed that carbon emissions from plantations are mainly due to changes in soil structure. They estimated agricultural carbon emissions based on crop characteristics, climatic conditions, and soils (Vleeshouwers and Verhagen, 2002) [8].

2.1.2. Factors Influencing Agricultural Carbon Emissions

Agricultural production activities have long cycles and are highly vulnerable to natural climate change, with high uncertainty. Its characteristics make its causes and influencing factors more complex, making its mechanism challenging to clear. To achieve the emission reduction target for agricultural products in China, we must first quantify and identify it. The related research on carbon emissions of agricultural products in China focuses on two aspects: first, the study of the mechanism of external conditions such as carbon reduction, employment policy, and innovation ability on carbon emissions of agricultural products in China.

Moreover, Xu et al. (2023) found that the carbon tax outperforms carbon emission trading for emission reductions [9]. Pamuk et al. (2014) [10] used the data of eight African

countries for the study. The main driving factor for agricultural carbon emissions is innovation capacity. In addition, agricultural investment, employment policies, and restructuring affect agricultural carbon emissions differently (Lei et al., 2017) [11]. Second, it is caused by internal reasons, such as the agricultural production mode. The mode of cultivated land is an essential factor affecting the carbon emissions of the farmland ecosystem in China, while under non-intensive, intensive, and traditional cultivation methods, the carbon emissions of the farmland ecosystem show an upward trend (Baumann et al., 2017) [12]. Changes in land use affect agricultural carbon emissions (Gamboa and Galicia, 2011) [13]. Cui et al. (2019) [14] pointed out that different intercropping practices also impact crop carbon emissions.

### 2.2. Studies on the Digital Economy

### 2.2.1. Measurement of the Digital Economy Development Index

Tapscott (1996) [15] first introduced the term "digital economy", and there is no single definition of the term in academic circles. The definition of the term has varied, and scholars have focused on three main areas: The first is digital information, i.e., stored images, text, sound, etc. (Berisha-Shaqiri and Berisha-Namani, 2015) [16]. The second category is the digital technology industry, including communications equipment manufacturing, information technology services, digital content industry, and e-commerce platforms (Colecchia et al., 2014) [17]. The third is digital technologies (Richter et al., 2017) [18]. Many organisations have developed digital economy indices. For example, the degree of digitisation of firms and industries, the degree of digitisation of economic activity and output, the combined impact of various economic indicators, and the monitoring of new digital sectors (Barefoot et al., 2018) [19]. The World Bank's Global Inclusive Finance Database (FINDEX) measures the extent of digital financial inclusion across dimensions such as universal account use, savings behaviour, credit behaviour, and insurance behaviour (Demirguc-Kunt, 2015) [20]. In addition, the ICT Development Index (IDI) of the International Communication Organisation and the Digital Economy and Society Indicator (DESI) of Europe are used to measure it on many levels. In China, the relevant departments have assessed the "digital economy", such as the Digital Economic Competitiveness Index (Digital Economics Index) of the Shanghai Academy of Social Sciences, the China Institute of Telecom Science and Technology (China Information Communications Institute of Digital Economics), and "Inclusive Finance in China" (referred to as "Digital inclusive Finance") published by Guo Yin and other scholars (2016) [21] Collaboration with the Digital Finance Research Center of Peking University.

### 2.2.2. Digital Economy and Agriculture

Countries worldwide have accumulated rich experience in using digital technology to promote agricultural and rural development. With the development of the Internet, the competitiveness, resource utilisation level, productivity level, and business efficiency of rural industries have improved (Hailu et al., 2014) [22]. The digital economy in rural areas reduces information asymmetry, expands the scale effect of agricultural production, optimises factor flow channels, reshapes the original factor allocation structure, improves allocation efficiency, increases the effective supply of labour, reduces transaction costs, and also improves the efficiency of risk management (Irwin et al., 2010; Goldfarb and Tucker, 2019) [23,24]. Moreover, the advantages of the digital economy, such as its renewable, non-competitive, inclusive, and non-exclusive nature, and its characteristics of high penetration, external economies, and incremental marginal benefits, allow for self-renewing iterations and generate multiplier effects (Bukht et al., 2018) [25]. At the same time, data intelligence can enhance the monitoring and analysis of agricultural greenhouse gas emission sources, leading to precise management and intervention to reduce carbon emissions (Henderson et al., 2020) [26].

### 2.2.3. Digital Economy and Carbon Emissions

Research shows that it is an effective way to alleviate environmental problems such as carbon dioxide through scientific and technological innovation (Brock and Taylor, 2010) [27], and in particular, innovations oriented towards the advancement of green technologies play an essential role in carbon reduction (Lee and Min, 2015) [28]. In the digital industrialisation phase, hardware manufacturing is highly energy-intensive, and digital economy industries such as telecommunications, I.T. services, and the Internet are highly power-intensive (Williams, 2011) [29]. Studies have also found that the digital economy is more agglomerative, with very intensive equipment and facility inputs, and that these facility devices require large amounts of electricity to operate (Salahuddin and Alam, 2015; Hittingere and Jaramillo, 2019) [30,31]. Research shows that with the development of the digital industry, carbon dioxide emissions in China are also increasing. Under this background, this project puts forward a new mode of new energy power generation based on new energy vehicles—the sustainable development model based on new energy vehicles (Anser et al., 2021) [32]. Digital technologies can also shorten clean energy R&D cycles, improve R&D efficiency, and enhance the immediate responsiveness of government (Allam and Jones, 2021) [33]. Digital governance theory has guided companies to improve pollution control capabilities and environmental optimisation (Fankhauser, 2013) [34]. Regarding $CO_2$ emissions, Li et al. (2021) [35] showed nonlinear characteristics of an inverted U shape between $CO_2$ and the digital economy.

### 2.3. Digital Technology Index System Construction and Measurement

Based on digital technology development, this paper constructs the Digital Technology Development Index from the development of digital industries and technology applications. The primary indicator of the digital economy development includes four secondary indicators. A secondary indicator for the first level of digital technology adoption: digital financial inclusion. The specific construction of each province's index of the level of digital technology development and the indicator weights are shown in Table 1.

**Table 1.** Digital technology index system.

|  | Primary indicators | Indicator weights | Secondary indicators | Indicator weights | Indicator attribute (+/−) |
|---|---|---|---|---|---|
| Digital technology index | The development of the digital industry | 0.904 | Percentage of employees in computer services and software | 0.187 | + |
|  |  |  | Total telecom services per capita | 0.359 | + |
|  |  |  | Mobile subscribers per 100 persons | 0.146 | + |
|  |  |  | Internet users per 100 persons | 0.212 | + |
|  | The application of digital technology | 0.096 | China's Digital Inclusive Finance Index | 0.096 | + |

The data above are from the Chinese Bureau of Statistics. As the core indicator of this paper, the construction of the digital technology development level index includes China's digital inclusive finance data (Guo et al., 2016) [21]. The entropy method is more scientific and objective in determining the weights. In information theory, entropy refers to the measurement of uncertainty. On this basis, a multi-objective decision making based on information entropy is proposed. The weight of each evaluation index is divided by that using the method, and each evaluation index is treated uniformly. The entropy method requires the inversion of negative indicators and the forwarding of positive ones. As the digital technology measurement indicator system in this paper does not contain negative indicators, it does not need to be reversed. Table 2 shows the average digital technology level.

**Table 2.** The average digital technology level in different regions of China from 2011 to 2019.

| Regions | 2011 | 2012 | 2013 | 2014 | 2015 | 2016 | 2017 | 2018 | 2019 | Average |
|---|---|---|---|---|---|---|---|---|---|---|
| Beijing | 0.333 | 0.371 | 0.415 | 0.447 | 0.506 | 0.573 | 0.532 | 0.531 | 0.557 | 0.4739 |
| Tianjin | 0.071 | 0.100 | 0.111 | 0.165 | 0.245 | 0.197 | 0.200 | 0.206 | 0.210 | 0.1672 |
| Hebei | 0.038 | 0.051 | 0.078 | 0.081 | 0.093 | 0.099 | 0.138 | 0.131 | 0.145 | 0.0949 |
| Shanxi | 0.046 | 0.063 | 0.089 | 0.089 | 0.101 | 0.114 | 0.120 | 0.130 | 0.158 | 0.1011 |
| Inner Mongolia | 0.069 | 0.077 | 0.107 | 0.104 | 0.116 | 0.122 | 0.134 | 0.154 | 0.165 | 0.1164 |
| Liaoning | 0.067 | 0.086 | 0.111 | 0.120 | 0.134 | 0.143 | 0.163 | 0.172 | 0.190 | 0.1318 |
| Jilin | 0.071 | 0.074 | 0.100 | 0.107 | 0.119 | 0.126 | 0.138 | 0.146 | 0.115 | 0.1107 |
| Heilongjiang | 0.056 | 0.063 | 0.140 | 0.099 | 0.110 | 0.118 | 0.138 | 0.151 | 0.164 | 0.1154 |
| Shanghai | 0.154 | 0.181 | 0.388 | 0.292 | 0.296 | 0.310 | 0.343 | 0.390 | 0.410 | 0.3071 |
| Jiangsu | 0.068 | 0.087 | 0.125 | 0.123 | 0.133 | 0.149 | 0.170 | 0.191 | 0.212 | 0.1398 |
| Zhejiang | 0.096 | 0.120 | 0.146 | 0.155 | 0.168 | 0.179 | 0.208 | 0.221 | 0.237 | 0.1700 |
| Anhui | 0.025 | 0.041 | 0.063 | 0.079 | 0.090 | 0.100 | 0.096 | 0.129 | 0.135 | 0.0842 |
| Fujian | 0.063 | 0.083 | 0.113 | 0.116 | 0.138 | 0.147 | 0.167 | 0.189 | 0.210 | 0.1362 |
| Jiangxi | 0.023 | 0.053 | 0.070 | 0.081 | 0.087 | 0.091 | 0.113 | 0.119 | 0.126 | 0.0848 |
| Shandong | 0.036 | 0.059 | 0.093 | 0.104 | 0.121 | 0.115 | 0.129 | 0.141 | 0.148 | 0.1051 |
| Henan | 0.016 | 0.033 | 0.056 | 0.064 | 0.078 | 0.089 | 0.102 | 0.122 | 0.169 | 0.0810 |
| Hubei | 0.039 | 0.060 | 0.079 | 0.089 | 0.098 | 0.106 | 0.123 | 0.154 | 0.164 | 0.1013 |
| Hunan | 0.037 | 0.054 | 0.069 | 0.081 | 0.089 | 0.092 | 0.105 | 0.118 | 0.130 | 0.0861 |
| Guangdong | 0.106 | 0.124 | 0.152 | 0.164 | 0.164 | 0.176 | 0.205 | 0.246 | 0.256 | 0.1770 |
| Guangxi | 0.042 | 0.062 | 0.072 | 0.069 | 0.079 | 0.123 | 0.145 | 0.185 | 0.197 | 0.1082 |
| Hainan | 0.126 | 0.138 | 0.160 | 0.188 | 0.148 | 0.163 | 0.179 | 0.217 | 0.214 | 0.1703 |
| Chongqing | 0.065 | 0.063 | 0.085 | 0.095 | 0.110 | 0.103 | 0.118 | 0.138 | 0.147 | 0.1027 |
| Sichuan | 0.034 | 0.062 | 0.095 | 0.108 | 0.127 | 0.170 | 0.185 | 0.198 | 0.179 | 0.1287 |
| Guizhou | 0.026 | 0.035 | 0.062 | 0.071 | 0.082 | 0.092 | 0.108 | 0.112 | 0.118 | 0.0784 |
| Yunnan | 0.034 | 0.050 | 0.086 | 0.085 | 0.090 | 0.098 | 0.106 | 0.125 | 0.140 | 0.0904 |
| Shanxi | 0.071 | 0.115 | 0.173 | 0.120 | 0.125 | 0.138 | 0.156 | 0.177 | 0.186 | 0.1401 |
| Gansu | 0.019 | 0.037 | 0.059 | 0.070 | 0.082 | 0.096 | 0.111 | 0.120 | 0.159 | 0.0837 |
| Qinghai | 0.073 | 0.105 | 0.087 | 0.078 | 0.124 | 0.265 | 0.295 | 0.327 | 0.437 | 0.1990 |
| Ningxia | 0.039 | 0.063 | 0.087 | 0.178 | 0.125 | 0.103 | 0.115 | 0.121 | 0.138 | 0.1077 |
| Xinjiang | 0.151 | 0.118 | 0.135 | 0.136 | 0.154 | 0.135 | 0.176 | 0.208 | 0.230 | 0.1603 |

From Table 2, China's digital economy development shows an upward trend, with all provinces having a higher level of digital economy development in 2019 than in 2011. Beijing has the highest average digital technology development level index, and the eastern coastal regions of Shanghai, Guangdong, Zhejiang, Hainan, Liaoning, and Fujian also have significantly higher levels of digital economy development than other regions do. That of the eastern region is higher than those of the central and western regions, that of the coastal region is higher than that of the inland region, and the "digital divide" is obvious. In 2011, the level of digital technology development was relatively low in most regions of China, except for Beijing, Shanghai, Guangdong, and Hainan, where the digital technology development level index was higher than 0.07. All other regions had a digital technology development level index of 0.01 to 0.07. Since the country officially launched the "digital economy" in 2016, digital technology in various parts of China has been improving continuously, and the development degree of digital technology in each region is more than 0.16. The degree of digital technology in Beijing is 0.5, while that in Shanghai is more than 0.3, which is much higher than those of all provinces and cities in China. Overall, China's gap is still significant, with a relatively clear regional digital divide in China.

Figure 1 shows the overall score and growth rate of digital technology development from 2011 to 2019 in China. China's digital technology development level was only 0.0698 in 2011, and the highest increase in the decade was in 2013 with 3.26%. China's digital technology development started late and did not form a stable growth trend until after 2015. Compared with the United States, which ranks first, the total size of China's digital economy in 2020 was less than 40 per cent of that of the United States (USD 13.6 trillion), and it is still catching up. Taking Table 2 and Figure 1 together, we can observe that although

the development of the digital economy varies from fast to slow in different provinces, there is generally less volatility, especially in the more economically developed regions such as Beijing, Shanghai, and Zhejiang, where the development of digital technology is at a higher level due to the earlier establishment of the infrastructure of the digital economy, and the more significant investment in it. The second gradient regions, such as Guizhou, Shaanxi, Hubei, Hunan, and Qinghai provinces, have gradually formed a new digital technology development pattern as the area of digital technology development has expanded.

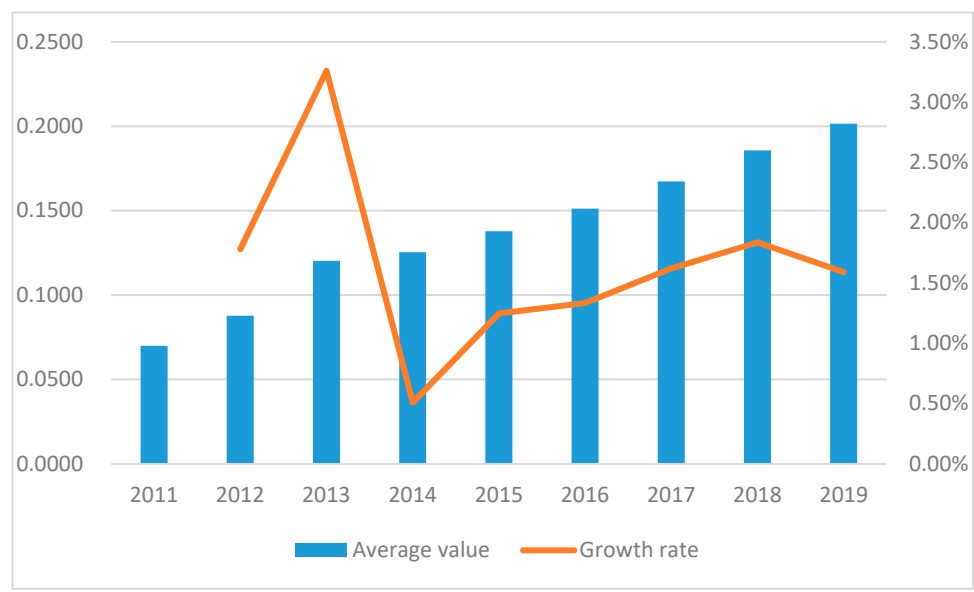

**Figure 1.** Digital technology development level and growth rate in China, 2011–2019.

## 3. Empirical Study of the Impact of Digital Technology on the Carbon Intensity of Agricultural Emissions

### 3.1. Model Setup

According to the purpose of this paper, the panel data regression model was constructed using agricultural carbon intensity (ACR) as the independent variable, digital technology (dig) as the core independent variable, and appropriate control variables (X) in the baseline model:

$$ACR_{i,t} = \alpha \cdot dig_{i,t} + \beta \cdot Z_{i,t} + \xi_i + \omega_{i,t} \tag{1}$$

In Equation (1), $ACR_{i,t}$ denotes the carbon intensity of agricultural emissions in region $i$ in year $t$, $dig_{i,t}$ denotes the level of digital technology in region i in year t, and Z denotes the control variables. The information is shown below in detail.

Dependent variable: Agricultural carbon intensity ($ACR$). There is currently no direct statistical measure of carbon emissions of agricultural products, and other indicators have to be used for estimation. Carbon dioxide emissions from agriculture come from two sources. The first is the use of fertilisers (mainly anthracite coal), pesticides, and plastic sheeting, which are also direct carbon emissions. The second is the indirect use of fossil fuels in agricultural production, such as tillage, irrigation, and transport, which uses diesel fuel to power farm machinery. Based on this, this paper refers to Zhu and Huo (2022) [36] for the estimation of China's agricultural carbon emissions, as shown in Equation (2).

$$car_{j,t} = \sum CO_{2,i} = \sum E_i \cdot \alpha_i \tag{2}$$

where $car_{j,t}$ is the carbon emissions of the $j-$ th region in year $t$, $CO_{2,t}$ is the carbon emissions of the $i-$ th production activities in the different regions, $E_i$ is the use of the carbon emission source of the i-th production activities, and $\alpha_i$ is the carbon emission factor of the $i-$ th carbon emission source, as shown in Table 3. As shown in Table 3, the main

production factors in China's agricultural production process are cultivated land, diesel oil, agricultural film, pesticides, chemical fertilisers, irrigation, etc. With the continuous global expansion, agricultural production contributes more and more to greenhouse gas emissions.

**Table 3.** Carbon emission factors and sources.

| Sources | Factors |
|---------|---------|
| Ploughing | 312.6 kg C/km$^2$ |
| Diesel | 0.5927 kg C/kg |
| Agricultural film | 5.18 kg C/kg |
| Pesticides | 4.934 kg C/kg |
| Fertiliser | 0.8956 kg C/kg |
| Irrigation | 25 kg C/km$^2$ |

Data source of carbon emission: China Agricultural University, Zhu and Huo (2022) [36], Dubey et al. (2009) [37].

Independent variable: digital technology (dig). In the previous section, this paper presents the measurement of digital technology for 30 regions in China from 2011 to 2019. It is not repeated here.

Control variables: ① The level of agricultural exportation is used to measure the level of international trade and openness of the local primary sector, as represented by the scale of agricultural export/total agricultural production. ② The damage to cultivated land. The most fundamental difference between agricultural production and other industries is greatly affected by the natural environment, so we choose the degree of agricultural loss as the measurement index, and its ratio to the area of farmland is used as an index to measure agricultural loss. ③ Urbanisation. The process of civilisation and the development of society have different effects on the production and lifestyle of the countryside. ④ Industrial structure is not only a process of modernisation that reflects a country but also a complete process of industrialisation. ⑤ The population income level, which represents the population's standard of living in agricultural production areas. ⑥ Financial support to agriculture, which is government financial expenditure on agriculture, forestry, and water, is calculated herein using pairs of this indicator.

### 3.2. Data Sources and Descriptive Statistical Analysis

In the data used for the empirical tests in this chapter, the core independent variable "digital technology" was measured as above. To eliminate the impact of heteroscedasticity and the quantity of related variables on the results, we performed logarithmic processing on the data of agricultural financial support. Table 4 presents the detailed information of all variables.

**Table 4.** Descriptive statistical analysis of variables.

| | Variable Name | Mean | Standard Deviation | Min | Max |
|---|---|---|---|---|---|
| Dependent variable | Agricultural carbon intensity | 0.3294 | 0.2011 | 0.0829 | 1.6522 |
| Independent variable | Digital technology | 0.1385 | 0.0917 | 0.0160 | 0.5728 |
| | Exports of agricultural products | 0.0585 | 0.0856 | 0.0031 | 0.4561 |
| | Extent of damage to arable land | 815.17 | 771.79 | 0.00 | 4223.70 |
| Control variables | Urbanisation level | 58.38 | 12.30 | 35.03 | 89.60 |
| | Industrial structure | 46.38 | 9.68 | 29.70 | 83.50 |
| | Income per person | 2.2997 | 1.0984 | 0.82 | 7.22 |
| | Financial support for agriculture (log) | 6.1194 | 0.5636 | 4.5190 | 7.1780 |

Data sources: Chinese Bureau of Statistics.

### 3.3. Benchmark Regression Results

Before running the basic regression, the model was first tested using Hausman's test, and the results showed a value of 0.006, rejecting the initial assumption that the disturbance

term is unrelated to individual characteristics. Therefore, the baseline regression model utilised fixed effects for the regression analysis. To ensure the rigour of the model, baseline regression was performed here by gradually adding variables through stepwise regression. The results are shown in Table 5.

**Table 5.** Benchmark regression results.

| | Model (1) FE | Model (2) FE | Model (3) FE | Model (4) FE | Model (5) FE | Model (6) RE |
|---|---|---|---|---|---|---|
| Digital technology | −0.9619 *** (−16.16) | −0.9392 *** (−15.51) | −0.5706 *** (−7.26) | −0.6581 *** (−7.20) | −0.6102 *** (−6.63) | −0.5596 ** (−2.23) |
| Exports of agricultural products | | 0.3499 ** (2.03) | 0.0264 (0.18) | −0.1618 (−0.91) | −0.1229 (−0.70) | −0.0169 (−0.08) |
| Extent of damage to arable land | | 0.0185 *** (2.97) | 0.0012 (0.23) | 0.0010 (0.19) | 0.0021 (0.39) | 0.0033 (0.59) |
| Urbanisation level | | | −0.0135 *** (−9.96) | −0.0143 *** (−10.09) | −0.0125 *** (−8.00) | 0.0043 ** (2.58) |
| Industrial structure | | | 0.0045 *** (4.74) | 0.0040 *** (4.07) | 0.0044 *** (4.54) | 0.0257 (1.09) |
| Income per person | | | | 0.0176 * (1.85) | 0.0257 ** (2.60) | −0.0841 ** (−2.31) |
| Financial support for agriculture | | | | | −0.0578 *** (−2.67) | 1.290 *** (5.44) |
| Constant | 0.4626 *** (52.97) | 0.4239 *** (30.28) | 0.9880 *** (17.36) | 1.042 *** (16.35) | 1.237 *** (12.80) | −7.442 *** (−3.86) |
| Prob > F | 0.0000 | 0.0000 | 0.0000 | 0.0000 | 0.0000 | 0.0000 |
| N | 270 | 270 | 270 | 270 | 270 | 270 |

Note: ***, **, and * indicate significance at the 1%, 5%, and 10% levels, respectively, as in the table below.

From Table 5, when the control variable is added, the absolute value of digital technology factors decreases, but they all have obvious positive significance at 1%. In model (1), the factor of digital technology is −0.9619. That is to say, after controlling other factors, the development of digital technology will reduce the greenhouse gas emissions of agricultural products by 0.9619 units. Without these control variables, it naturally does not matter. In model 5, the factor related to science and technology is −0.6102, which is significant. After the increase in control variables, the indicators of digital technology decrease to a certain extent, but we can see that the impact of the development of digital technology on carbon dioxide is very significant and stable.

Regarding the control variables, the coefficients of urbanisation level, industrial structure, per-capita income of residents, and financial support for agriculture in model (5) are −0.0125, 0.0044, 0.0257, and −0.0578, respectively. The regression results of urbanisation level, industrial structure, and financial support for agriculture on the carbon emission intensity of agriculture are all significant at the 1% level. In terms of coefficients, urbanisation will reduce the carbon emission intensity of agriculture. This is because urbanisation will, on the one hand, promote the large-scale and intensive use of arable land. On the other hand, it will also introduce advanced urban technologies into the countryside, promoting the reduction in agricultural carbon emissions. Financial support for agriculture has a strong inhibiting effect on the carbon intensity of agriculture. The largest obstacle to the smooth and healthy development of agriculture is the lack of financial investment. Once sufficient funds are available to improve production technology and increase production efficiency, the intensity of carbon emissions will be reduced. The upgrade in industrial structure and the increase in per-capita income will lead to an increase in the carbon emission intensity of agriculture. Especially in the early stages of upgrading industrial structures, the tertiary industry's development will squeeze the primary industry's capital and labour input to a certain extent. As the population's income level rises, the consumption of agricultural products will be upgraded, shifting from grain consumption to higher meat and protein consumption, which will naturally lead to an increase in carbon emission intensity.

### 3.4. Heterogeneity Test

The benchmark regression model established in the previous part shows that digital technology can effectively reduce the carbon emissions of agricultural products. Because China has a vast territory, different regions have obvious differences in digital technology, agricultural development, industrial structure, resources, the environment, and other factors. For our research results to be more scientific, we conducted a heterogeneity test on the impact of digital technology on the intensity of agricultural carbon emissions. The results are shown in Table 6.

**Table 6.** Regression results for eastern, central, and western regions (fixed effects).

| Variables | Eastern Region Model (7) | Central Region Model (8) | Western Region Model (9) |
|---|---|---|---|
| Digital technology | −0.2284 *** | −0.6069 *** | −0.8809 *** |
| | (−4.52) | (−3.14) | (−5.28) |
| Exports of agricultural products | −0.0324 | 3.5906 *** | 0.5826 |
| | (−0.48) | (3.89) | (0.49) |
| Extent of damage to arable land | −0.0060 | 0.0004 | 0.0033 |
| | (−1.02) | (0.11) | (0.19) |
| Urbanisation level | −0.0068 *** | −0.0100 *** | 0.0002 *** |
| | (−6.14) | (−2.88) | (0.04) |
| Industrial structure | −0.0005 | 0.0020* | 0.0098 *** |
| | (−0.68) | (1.77) | (3.96) |
| Income per person | −0.0205 *** | −0.0076 | −0.0583 |
| | (−3.16) | (−0.32) | (−0.97) |
| Financial support for agriculture | −0.0228 * | 0.0275 | −0.2191 *** |
| | (−1.83) | (0.95) | (−3.52) |
| Constant | 0.7555 *** | 0.6486 *** | 1.5228 *** |
| | (12.81) | (3.75) | (5.49) |
| Provincial Effect | Yes | Yes | Yes |
| Year Effect | Yes | Yes | Yes |
| Prob > F | 0.0000 | 0.0000 | 0.0000 |
| N | 99 | 90 | 81 |

Note: *** and * indicate significance at the 1% and 10% levels.

Table 6 shows all inhibitory benefits, but the magnitude of the impact differs, with regression coefficients of −0.2284, −0.6069, and −0.8809, respectively. This is mainly because of the 13 major grain-producing areas in China, so from a regional point of view, China's grain-producing areas are relatively scattered. However, China has a vast territory and uneven inter-regional development (Luo and Chen, 2023) [38].

### 3.5. Robustness Tests

Natural endowments, the agricultural economic environment, and changes in development stages all influenced agricultural carbon emissions intensity. Therefore, to ensure the rigour of our empirical evidence, we next re-tested the robustness of the empirical evidence by substituting the dependent variable, shrinking the tails, and performing a brief analysis (Table 7).

① Replacement of dependent variable. In the fundamental regression analysis, the carbon dioxide emissions of agriculture are taken as the explained variable. Therefore, we took agricultural production capacity as the dependent variable to test the benchmark regression's robustness. In Table 7, the coefficients of digital technology in model (10) and model (11) are −0.8173 and −0.9675, respectively.

② Replacement of independent variable. The Index of Integration Development (IOD) is an indicator used to quantify the overall integration of industrialisation and informatization in the region. The current Regional Integration Development Index (IOD) is a quantitative result obtained via the dimensionless processing of the three-level indicators by taking logarithms and weighting them according to the weight of each level. To some

extent, this can represent the level of digital technology development in a region, so this indicator is used in this paper to replace the independent variable and to test the robustness of the model above. The coefficients of the two integration development indices in model (12) and model (13) are −0.0249 and −0.0238, respectively, consistent with the baseline regression results.

③ Tail shrinkage. Given the impact that extremes in the data have on robustness, in this paper, we applied bilateral 2% and 5% tail shrinkage to all continuous variables to further mitigate the effect of extreme values. After the tailing process, the regression results show little difference from the original results.

**Table 7.** Robustness tests.

| | Replacing the Dependent Variable | | Replacing Independent Variable (fii) | | Tailoring | |
| | Model (10) | Model (11) | Model (12) | Model (13) | Model (14) | Model (15) |
| | FE | RE | FE | RE | FE | RE |
|---|---|---|---|---|---|---|
| Digital Inclusion Index | | | −0.0249 *** | −0.0238 *** | | |
| | | | (−2.91) | (−3.38) | | |
| Digital technology | −0.8173 * | −0.9675 *** | | | −0.4351 *** | −0.4026 *** |
| | (−1.81) | (−2.96) | | | (−6.26) | (−2.64) |
| Control variables | YES | YES | YES | YES | YES | YES |
| Constant | 7.248 *** | 6.929 *** | 1.216 *** | 1.258 *** | 1.103 *** | 1.139 *** |
| | (15.22) | (6.23) | (10.46) | (4.56) | (15.38) | (7.33) |

Note: *** and * indicate significance at the 1% and 10% levels.

## 4. Conclusions, Policy Recommendations, and Research Outlook

### 4.1. Conclusions

Digital technology indicators enable quantitative research to be performed on it using the information entropy method. The results of this paper are summarised below.

First, on the whole, China still has a lot of room for progress in digital technology. Although the overall development momentum is still strong, the regional digital gap in China is still considerable.

Second, digital technology has a significant effect on reducing carbon dioxide emissions of agricultural products, and it is very stable. The higher the development of electronic technology, the smaller its greenhouse gas emissions. Urbanisation and rural financial support are conducive to reducing carbon emissions in rural areas of our country.

### 4.2. Policy Recommendations

Our findings have the following essential policy implications.

Firstly, we should accelerate the construction of hardware facilities for digital technology in rural areas. China is a vast country with relatively scattered agricultural resources and uneven agricultural development, especially in less developed regions such as southwest and northwest China, wherein infrastructural development is relatively poor. The Chinese government should accelerate the construction of communication facilities such as base stations and broadband in rural areas, especially in underdeveloped areas, to maximise the coverage of the agricultural digital economy.

Secondly, we should increase the promotion and popularisation of knowledge of the agricultural digital economy to promote the deep integration of the digital economy and agriculture. China still mainly relies on small-scale agriculture, with a small production scale, scattered farmers, and significant regional differences. The inherent national conditions of Chinese agriculture result in the slow and inefficient transfer of information regarding advanced agricultural management knowledge and technology. With the implementation of the "Digital Rural" strategy, digital technology development has accelerated in rural areas, and the popularisation of technologies such as the Internet and mobile phones has broken temporal and spatial boundaries of information dissemination. The government should use the Internet, communication equipment, and other means to promote and popularise advanced agricultural knowledge and technology and improve the awareness and application

of the digital economy and knowledge of carbon emissions among agricultural personnel. In addition, AI can help farmers better understand and control their energy use to optimise, optimise, and reduce energy consumption and emissions for sustainable development.

Finally, we should formulate different regional policies according to local conditions to promote digital technology application in agricultural production. The central and western regions have large primary industries, low levels of digital technology development, and a shortage of digital technology talent. Therefore, on the one hand, the government must invest significant funds and formulate appropriate policies to introduce agricultural digital technology talents from the eastern region. On the other hand, governments in the central and western regions need to encourage digital technology to shift towards agricultural production to safeguard agricultural carbon emissions.

### 4.3. Research Prospects

There are some limitations to this paper due to data collection constraints. On the one hand, due to data limitations, rural digital technology cannot be separated from the development of digital technology in the region as a whole. More research is needed to find more appropriate methods of measuring the level of rural digital technology. On the other hand, data are unavailable, making it difficult to accurately identify the causal effects of digital technology applications on residential agricultural carbon emissions. Future research could expand and deepen in these two directions.

**Author Contributions:** Conceptualization, Y.Z., X.W. and G.Z.; Methodology, Y.Z., X.W. and G.Z.; Validation, Y.Z., X.W. and G.Z.; Formal Analysis, Y.Z., X.W. and G.Z.; Data Curation, Y.Z., X.W. and G.Z.; Writing—Original Draft Preparation, Y.Z. and G.Z.; Writing—Review and Editing, Y.Z., X.W. and G.Z.; Visualization, Y.Z. and G.Z. Supervision, Y.Z.; Funding Acquisition, Y.Z. All authors have read and agreed to the published version of the manuscript.

**Funding:** Education Department of Hunan Supported Research Project (grant number: 21A0222; 20K004).

**Institutional Review Board Statement:** Not applicable.

**Informed Consent Statement:** Not applicable.

**Data Availability Statement:** No new data were created or analyzed in this study. Data sharing is not applicable to this article.

**Conflicts of Interest:** The authors declare no conflict of interest.

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
