# Peer review of "Blessing or Curse? The Impact of Digital Technologies on Carbon Efficiency in the Agricultural Sector of China"

_sustainability, doi:10.3390/su152115613_

Round 1
Reviewer 1 Report
Comments and Suggestions for Authors
Review points:
The article is interesting, however, as highlighted, there is still a need to insert some essential elements that make up a scientific article:
1. What are the potential implications of these findings for sustainable rural development and environmental conservation efforts?
2. The article presents considerable flaws in methodological strategies. The details of the research steps that were taken are missing.
3. What search protocol was followed?
4. What was the process by which the considered articles were collected? What are the inclusion and exclusion criteria?
5. In which period of time was the data analyzed?
6. How does digital technology contribute to reducing carbon emissions in the agricultural sector?
7. What are the time-evolving characteristics analyzed in this paper
8. There is space for improvement in the narratives and English use.
Comments on the Quality of English Language
1. There is space for improvement in the narratives and English use.
2. Many grammatical errors need to be rectified
Reviewer 2 Report
Comments and Suggestions for Authors
This study is indeed very meaningful and valuable, but there are still some mistakes that need to be noted:
1. The description in the table should be standardized.
2. The format of literature should be unified, and the proportion of literature in recent five years is too low, which needs to be increased
Comments on the Quality of English LanguageNo
Reviewer 3 Report
Comments and Suggestions for Authors
enclosed

needs improvement
Reviewer 4 Report
Comments and Suggestions for Authors
Comments to Authors
Dear Authors
I have reviewed this article and have some suggestions for you. The article presents a comprehensive study of the work done in the field of digital technology on carbon efficiency on the agricultural sector.
So in the introduction section I will suggest you to include some global scenario also where impact of digital technology is impacting agricultural sector. Currently I have felt that it more concentrated with China only.
In the table provide reference to the primary indicator in the table itself. I have a further concern about the Data Source: How were the Chinese provincial panel data from 2011 to 2019 obtained? Are there any concerns about the accuracy or comprehensiveness of this data?
Can you discuss the potential limitations of the study? Are there any assumptions or constraints that readers should be aware of?
Why are the main factor where the suppression effect be greater in the western region than in the central and eastern regions? Are there specific socio-economic or technological reasons behind this?
Finally, how relevant is this research in today's context, especially in relation to global challenges like climate change and sustainable development?

English is fine only minor correction required
